# Pretreatment Cancer-Related Cognitive Impairment—Mechanisms and Outlook

**DOI:** 10.3390/cancers11050687

**Published:** 2019-05-16

**Authors:** Brennan Olson, Daniel L. Marks

**Affiliations:** 1Medical Scientist Training Program, Oregon Health & Science University, 3181 SW Sam Jackson Park Rd, Portland, OR 97239, USA; olsobr@ohsu.edu; 2Papé Family Pediatric Research Institute, Oregon Health & Science University, 3181 SW Sam Jackson Park Rd, Portland, OR 97239, USA; 3Brenden-Colson Center for Pancreatic Care, Oregon Health & Science University, 3181 SW Sam Jackson Park Rd, Portland, OR 97239, USA

**Keywords:** Cancer, cachexia, cognitive decline, cytokines, extracellular vesicles, blood-brain barrier, neuroinflammation

## Abstract

Cognitive changes are common in patients with active cancer and during its remission. This has largely been blamed on therapy-related toxicities and diagnosis-related stress, with little attention paid to the biological impact of cancer itself. A plethora of clinical studies demonstrates that cancer patients experience cognitive impairment during and after treatment. However, recent studies show that a significant portion of patients with non-central nervous system (CNS) tumors experience cognitive decline *prior* to treatment, suggesting a role for tumor-derived factors in modulating cognition and behavior. Cancer-related cognitive impairment (CRCI) negatively impacts a patient’s quality of life, reduces occupational and social functioning, and increases morbidity and mortality. Furthermore, patients with cancer cachexia frequently experience a stark neurocognitive decline, suggesting peripheral tumors exert an enduring toll on the brain during this chronic paraneoplastic syndrome. However, the scarcity of research on cognitive impairment in non-CNS cancers makes it difficult to isolate psychosocial, genetic, behavioral, and pathophysiological factors in CRCI. Furthermore, clinical models of CRCI are frequently confounded by complicated drug regimens that inherently affect neurocognitive processes. The severity of CRCI varies considerably amongst patients and highlights its multifactorial nature. Untangling the biological aspects of CRCI from genetic, psychosocial, and behavioral factors is non-trivial, yet vital in understanding the pathogenesis of CRCI and discovering means for therapeutic intervention. Recent evidence demonstrating the ability of peripheral tumors to alter CNS pathways in murine models is compelling, and it allows researchers to isolate the underlying biological mechanisms from the confounding psychosocial stressors found in the clinic. This review summarizes the state of the science of CRCI independent of treatment and focuses on biological mechanisms in which peripheral cancers modulate the CNS.

## 1. Introduction

Advances in cancer treatments have greatly improved the overall survival of patients. However, the use of cancer therapies, including radiation, chemotherapy, and immunotherapies, are commonly associated with toxicities. A multitude of studies demonstrates that patients treated with systemic therapy or radiotherapy experience neurocognitive decline [1,2,3,4]. However, a new wave of research demonstrates that even prior to treatment, a significant portion of patients with non-CNS malignancies experience cognitive impairment [5,6]. Furthermore, patients with cancer cachexia frequently experience cognitive decline, suggesting tumor factors chronically affect cognition throughout the disease course. With 1.7 million new cases of cancer in 2018 in the US alone and due to an aging population, the prevalence of cancer is on the rise as is the population of patients in remission or cured of disease [7]. Consequently, as patients continue to live longer after their cancer is cured, the focus is shifting towards understanding long-term sequelae caused by cancer therapeutics or surgery, and special attention to the quality of life measurements has become paramount for patient care. While much of the focus is centered around mitigating therapy-related toxicities, there is a paradoxical under-appreciation of inherent cancer-related toxicities. The precise mechanisms by which peripheral tumors communicate to the CNS to cause cognitive and behavioral change remain elusive, but almost certainly consist of a complex interplay between the immune system, genetic factors, host behavior, and psychosocial state. 

Overall, an estimated 30% of cancer patients have detectable cognitive impairment prior to treatment, up to 75% of patients experience cognitive impairment during treatment, and 35% of patients experience cognitive impairment several years after the completion of treatment [1,2,3,4,5,6]. Defining and measuring cancer-related cognitive impairment (CRCI) is a challenge, as there are multiple neuropsychological measures to assess cognitive function. Testing cognition within the clinic is complicated and measures multiple domains, including executive function, visual memory, psychomotor speed, attention, and concentration, to name a few. There is currently no “gold standard” for measuring cognitive function in cancer patients or survivors, and there is no general consensus on the methodology for the testing of cognitive decline [8]. The clinical presentation of CRCI varies between cancer types and is affected by several factors of the tumor alone, including the stage during diagnosis. As we review the recent clinical data of CRCI, it is worth noting that the clinical tests of cognitive dysfunction in cancer are not currently validated for this patient cohort. Furthermore, the definition of “cognitive impairment” encompasses several aspects of cognitive function and is not universal across studies. In addition to cognitive impairment, patients may also experience cancer-related fatigue prior to treatment [9,10]. Although out of the scope of this review, it is worth noting that cancer-related fatigue—defined as a persistent sense of tiredness related to cancer or cancer treatment that interferes with usual functioning—is biologically distinct from cognitive dysfunction [11,12]. While the neurocognitive decline may occur in the context of cancer-related fatigue, CRCI frequently occurs in isolation or precedes fatigue [13,14,15]. Additionally, the clinical course of fatigue is a notable experience that impairs both mind and body, while CRCI is often insidious in nature, and patients may not universally recognize their cognitive decline during clinical presentation [16,17]. Furthermore, defining cognitive decline requires knowledge of patients’ baseline cognitive function. Rodent models confirmed the temporal distinction between cognitive impairment, fatigue, and other sickness behaviors. Memory impairment, as measured by novel object recognition, significantly precedes fatigue and other classical sickness behaviors [18]. Indeed, repeated intraperitoneal injection of breast cancer cell-conditioned medium is sufficient to induce cognitive decline, but not other sickness behaviors [18], providing direct evidence of the ability of soluble tumor factors to cause cognitive impairment. 

In patients, fatigue is associated with poor clinical outcomes and is taken into consideration when creating treatment plans [19,20]. However, less was known about the prognostic utility of CRCI until recently. Cognitive impairment (deficits in working memory) prior to cancer treatment is an important predictor of survival in hematologic cancers [21] and pancreatic cancer [22]. Furthermore, CRCI is not limited to older patients, as it is also observed in younger patients with Hodgkin lymphoma [23]. Taken together, recent clinical studies demonstrate that cognitive impairment is an underappreciated symptom of cancer that affects both quality of life as well as survival. Furthermore, these studies emphasize the need for robust clinical testing of cognitive impairment in patients recently diagnosed with cancer, as CRCI is a risk factor that should guide clinicians in creating individualized treatment plans.

As the population continues to age, the prevalence of cognitive impairment in cancer patients will continue to escalate, creating a serious need for targeted approaches for patients with CRCI. A recent article by Horowitz et al. highlights the importance of future studies to investigate CRCI in a basic science context, efforts that are requisite for developing rational therapeutic targets [24]. As patients experience varying degrees of CRCI throughout their disease course, assessing the differential impact of the tumor and treatment on cognition remains challenging. A better pathophysiological understanding of CRCI will aid in the development of effective treatment strategies while providing a broader framework for mechanisms by which peripheral cancers communicate to the brain to modulate behavior. Consequently, animal models are necessary to uncouple tumor versus treatment effects on the CNS. This review summarizes recent clinical studies concerning CRCI and highlights advances in our understanding of the biological mechanisms by which peripheral tumors modulate the CNS.

## 2. Epidemiology of Pre-Treatment CRCI and Recent Clinical Studies

Cognitive impairment during cancer is most studied in the context of therapy-related cognitive impairment. However, pre-treatment CRCI was recently demonstrated in several cancer types, including acute myelogenous leukemia or myelodysplastic syndrome [11], breast cancer [5,25,26], colorectal cancer [27], and testicular cancer [28] (Table 1). Longitudinal studies observed between 11 [29] and 33% of breast cancer patients experienced cognitive impairment prior to chemotherapy [5,6,30]. More recently, a nationwide, prospective, observational study identified that patients with stage I-IIIC breast cancer experience significant cognitive impairment prior to treatment, particularly in the domains of memory, attention, and executive function [31].

In the past 25 years, several studies explored neuroimaging approaches in assessing cognitive impairment during cancer [32]. A recent magnetic resonance imaging (MRI) study in patients with various non-CNS cancers noted significant differences in cortical surface area or cortical thickness in multiple brain regions between non-treated cancer patients and controls [33]. Specifically, decreases in cortical surface area or thickness were observed in untreated cancer patients in the temporal and frontal lobes, including the parahippocampal region, an area important in memory encoding and retrieval. Scherling and colleagues also observed smaller white matter volumes in breast cancer patients compared to healthy controls in several regions of the parietal, frontal, and limbic regions [34]. These two studies provide compelling data describing baseline neuroanatomic differences, independent of treatment, between cancer patients and healthy controls. However, contrary results were reported in breast cancer brain MRI studies, revealing only a chemotherapy-related reduction in a gray matter [35,36]. It is worth noting that the atrophy observed in these studies was not directly associated with cognitive impairment. Importantly, these studies are not without limitations, as the sample size was limited and baseline performance status varied considerably amongst groups.

The notion that pre-treatment cancer patients experience neurocognitive decline more readily than age-matched controls is gaining attention. Furthermore, recent clinical studies demonstrate that cognitive decline prior to cancer treatment is a significant independent predictor of survival [21,22]. Taken together, the clinical studies to date suggest that peripheral tumors and blood cancers affect several domains of cognitive function and modulate white matter neuroanatomy. However, the mechanisms by which cancer interfaces with the brain to enact cognitive and structural change remain unclear. Here, we explore the putative mechanisms by which peripheral cancers interact with the brain to affect cognitive function.

## 3. Theory of Soluble Tumor Factors’ Ability to Communicate with the CNS

### 3.1. Inflammatory Cytokines

Perhaps the most established and studied facet of cancer and behavior is the inflammatory capacity of the tumor microenvironment. Tumors contain a complex network of tissue stroma, infiltrating immune cells, and neoplastic cells, all capable of generating cytokines [37,38,39,40,41]. Several peripheral cancers are associated with increased circulating inflammatory cytokines, indicating this phenomenology is not specific to any one cancer [42,43,44,45,46,47,48,49,50]. Furthermore, unique serum cytokine profiles were identified during neurocognitive decline in cancer: in patients with acute myelogenous leukemia or myelodysplastic syndrome, increased interleukin (IL)-6 was associated with poor executive function, while concurrent increases in serum IL-6, IL-1 receptor, and tumor necrosis factor (TNF)-alpha were associated with fatigue [11]. Interestingly, IL-8 levels positively correlated with memory function, suggesting that not all pro-inflammatory cytokines are deleterious to the CNS during cancer. Indeed, one study of breast cancer patients demonstrated that IL-17 and Granulocyte-colony stimulating factor (G-CSF) were positively correlated with psychomotor speed prior to treatment [51]. While several clinical studies have demonstrated that peripherally-derived cytokines correlate with cognitive decline [52,53,54,55], the precise biological mechanisms of cytokine-mediated communication with the CNS to affect cognition are still being unveiled.

Inflammatory cytokines produced within the tumor environment enter the circulation and interface with the brain by directly crossing the blood-brain barrier (BBB) [56] or through direct blood-borne sampling by circumventricular organs (CVOs) [57,58,59]. After crossing the BBB or highly-permeable capillaries of the CVOs, these inflammatory cytokines activate glial cells [60,61] or directly influence neuronal circuits important in homeostatic functions and cognition [62,63,64]. However, even before entering the brain parenchyma, inflammatory cytokines within the choroid plexus and meningeal vessels are capable of activating quiescent perivascular macrophages [65] and dendritic cells [66]. Activation of these antigen-presenting cells promotes further inflammatory response by recruiting cytokine-producing adaptive immune cells [67]. For instance, CNS-infiltrating T effector cells are especially prevalent in multiple sclerosis and secrete interferon (IFN)-gamma and IL-17A [68], cytokines that induce glial cell activation and apoptosis of myelin-producing oligodendrocytes [69]. Xiaoyu and colleagues recently demonstrated that brain-endothelial expression of the interleukin-1 receptor (IL-1R) enhances leukocyte recruitment, mediates sickness behavior, and impairs neurogenesis [61]. Indeed, brain endothelial cells respond to IL-1B in a myeloid differentiation primary response protein MyD88-dependent manner, which amplifies and propagates inflammatory signals to glial cells [70]. These studies demonstrate that the BBB serves to sense, interpret, and amplify peripheral cues to the CNS [70], potentially through further production of inflammatory cytokines at the BBB. During the progression of cancer, the cytokine array produced by both distant intratumoral cells and CNS-invading immune cells presents a unique challenge to the brain. The constant production of cytokines activates meningeal and choroid plexus immune cells, promotes inflammatory glial polarization, and ultimately inflicts pathological structural and biochemical changes in neuron populations central to cognitive function. Indeed, neuroinflammation caused by circulating cytokines during cancer progression parallels the emerging story of inflammation in primary neurodegenerative diseases [71].

Peripheral cytokines also mediate behavioral responses by influencing the vagus nerve. Vagal afferent fibers richly innervate peripheral organs and relay inflammatory, satiety, and metabolic cues to the brain. In the brainstem, these afferent fibers terminate in the nucleus tractus solitarius (NTS), which then sends fibers projecting to regions important in behavior and cognition. These regions include the locus coeruleus, hypothalamus, thalamus, and amygdala [72]. Electrophysiological recordings of the vagus nerve following peripheral IL-1B or TNF-alpha administration demonstrate that these pro-inflammatory cytokines robustly induce nerve activation [73]. Although no direct evidence exists connecting vagal nuclei to hippocampal structures, damage to the subdiaphragmatic vagus in rats results in microgliosis and impaired neurogenesis in the dentate gyrus [74]. While there is a large body of literature suggesting that direct vagal nerve stimulation (VNS) may improve cognitive function in rodent models of CNS disease [75,76], stimulation of the vagus nerve in these models is intermittent and independent of a systemic inflammatory milieu. It is conceivable that continuous vagal stimulation via the inflammatory tumor microenvironment is an exhaustive process on CNS circuits, while irregular stimulation through VNS treatment “jump starts” neurons during depression and neurodegenerative diseases.

Murine studies involving the injection of single cytokines into the periphery demonstrate the significance of peripheral inflammation on cognitive processes. Here, we review the literature of cytokines that are dysregulated during cancer and mechanisms by which these individual molecules alter cognition. 

#### 3.1.1. IL-1B

IL-1B remains one of the most studied cytokines during inflammatory diseases, yet direct connections between IL-1B and cognition lack, particularly in a behavioral neuroscience context. Intraperitoneal administration of IL-1B alone leads to cerebral release and catabolism of serotonin, norepinephrine [77], dopamine [78], and tryptophan [79]—vital neurotransmitters and molecules in cognitive processes. Indeed, brief exposure of cornu ammonis (CA1) pyramidal neurons to sub-femtomolar concentrations of IL-1B causes profound decreases in excitatory glutamatergic transmission, yet leaves inhibitory gamma-aminobutyric acid (GABA)ergic signaling unabated [80]. Furthermore, IL-1B and TNF-alpha can directly inhibit the ability of astrocytes to convert excess glutamate to glutamine, a process critical in the prevention of neuron excitotoxicity [81].

#### 3.1.2. IL-2

IL-2 is implicated in hippocampal neurodegeneration, as repeated peripheral injections in rats suppress long-term potentiation [82]. Mechanistically, IL-2 inhibits the hippocampal release of acetylcholine [83], a neurotransmitter important in attention, recall, and memory formation [84]. Axonal degeneration, demyelination, and alterations in the cerebrovascular structure are observed as early as 6 hours after peripheral IL-2 administration in rats [85]. These neuron structural changes persist and compound after repeated injections. IL-2 is a Food and Drug Administration (FDA)-approved treatment for metastatic melanoma and renal cell carcinoma, and it is associated with psychiatric manifestations and cognitive decline. Denicoff et al. demonstrated that 22 of 44 patients receiving IL-2 therapy became disoriented over the course of treatment and demonstrated cognitive deterioration [86]. Cognitive decline and other neuropsychiatric symptoms were dose- and time-dependent, as patients treated with higher doses presented with symptoms earlier, while nearly all patients exhibited neurologic symptoms at the end of the study. During the progression of CRCI, the secretion of IL-2 by intratumoral dendritic cells, CD4+ T cells, and CD8+ T cells aids in the immunologic response against the tumor. However, the rapid and sustained release of tumoral IL-2 results in spillage of the cytokine into circulation and migration to the CNS, potentially resulting in cognitive deficits.

#### 3.1.3. IL-6

IL-6 is recognized as a versatile cytokine in various CNS processes, proving to be beneficial for normal CNS physiology, yet deleterious during several neurological diseases [87]. Despite being classified as a pro-inflammatory cytokine, nearly 30 years ago evidence emerged demonstrating that IL-6 is a neurotrophic factor [88]. Indeed, several studies implicate IL-6 in neuronal differentiation [89,90,91]. However, IL-6 contributes to chronic neuropathologies by enhancing N-methyl-D-aspartic acid (NMDA) receptor neurotoxicity [92]. IL-6 is also capable of indirectly modulating neuronal activity by differentially activating microglia; IL-6-stimulated microglia co-cultured with neurons drastically reduce neuronal survival [93]. This translates behaviorally, as overexpression of IL-6 within astrocytes results in impaired avoidance learning behavior in mice [94]. Conversely, mice deficient in IL-6 display sensory dysfunction and impaired axonal regenerative capacity [95]. While the literature is full of conflicting evidence for the role of IL-6 in normal neuronal function and pathologies, it is clear that a delicate balance of IL-6 exists during basal physiology, while states of inflammation result in IL-6 aberrations that subdue the CNS and cause cognitive decline.

#### 3.1.4. TNF-alpha

Peripheral TNF-alpha is robustly induced during several peripheral inflammatory diseases, including cancer [96,97,98,99], and the significance of TNF-alpha signaling in the brain on cognitive function has gained attention in recent years. Terrando and colleagues demonstrated that TNF-alpha is an upstream target of IL-1B production in the brain, synergizes with MyD88 signaling, and sustains cognitive decline after surgical intervention in mice [100]. When TNF-alpha crosses the BBB, it can activate the astrocytic TNF-alpha receptor 1 (TNFR1), resulting in hippocampal synaptic alterations and subsequent memory impairment [101].

#### 3.1.5. Interferon

The interferons (IFN) provide a compelling human example in which single cytokines in the periphery can cause cognitive dysfunction. IFN-alpha, beta, and gamma are all FDA approved drugs used to treat cancer, hepatitis, multiple sclerosis, and chronic granulomatous disease. Patients treated with IFN-alpha, beta, or gamma routinely exhibit depression, disordered sleep, malaise, and impaired memory [102]. The molecular cascade in which IFN-B regulates behavior and cognitive impairment was recently unveiled by Blank and colleagues: circulating IFN-B binds to luminal brain endothelial IFN receptor chain 1 and mediates the release of C-X-C motif ligand 10 (CXCL10) into the brain parenchyma, which, in turn, causes neuronal dysfunction [103]. Furthermore, there is a strong correlation between prefrontal cortex (PFC) levels of CXCL10 and cognitive decline in humans [104]. The PFC is uniquely positioned to orchestrate several facets of cognition through its extensive neural network [105], and its activation is significantly reduced in breast cancer survivors irrespective of treatment [106]. During cancer, it is possible that the peripheral up-regulation of IFN-B induces CXCL10 expression in cerebral vessels of the PFC in a chronic fashion, ultimately leading to dysfunction of PFC circuits critical in cognitive processes.

Recent studies have greatly improved our understanding of the tumor-brain axis in the context of inflammation and cytokine release. While it is clear that cytokines play a central role in the normal development of the CNS [107,108], it is also apparent that the marked and sustained production of cytokines during the progression of cancer exerts negative consequences on the brain, including cognitive decline.

### 3.2. Brain-Infiltrating Immune Cells

Although the CNS was classically described as an immune-privileged organ, it is now accepted that peripheral immune cells infiltrate the brain during pathologic states. The early work characterizing brain-infiltrating immune cells was in the context of infectious disease, and a specific requirement for CNS immune cells during infection exists for controlling pathogen proliferation [109]. Although these brain immune cells limit pathogen proliferation, the CNS immunologic response is not universally beneficial when prolonged or exaggerated. Specifically, activated immune cells in the CNS cause inflammation and swelling of CNS structures and meninges. The collateral cerebral damage alters neuronal circuits to cause changes in behavior and cognition. While the role of infiltrating immune cells during CNS infections is well characterized, only recently their role has been explored in non-infectious diseases.

Over the past 15 years, various studies demonstrated that neuroinflammation secondary to non-infectious pathology leads to robust infiltration of peripheral immune cells into the brain. Conditions in which this phenomenon was observed include epilepsy [110], Alzheimer’s disease [111], Parkinson disease [112], multiple sclerosis (MS) [113,114], stroke [115,116], and pancreatic cancer [117]. However, the biological impact of these invading leukocytes during non-infectious disease has been unrecognized until recently. Several studies demonstrate mitigated brain inflammation, reduced disease progression, and improved functional outcomes by blocking leukocyte infiltration into the brain [111,118,119]. One such study demonstrated that preventing the entry of myeloid cells via chemokine receptor type 2 (CCR2) blockade was protective in a status-epilepticus murine model [118]. These studies demonstrated the close correlation between CNS disease progression and immune cell infiltrate. The identity of infiltrating immune cells in the CNS is heterogeneous and context-dependent, with varying immune cells populations within distinct regions of the CNS [120]. Nonetheless, it is clear that infiltrating immune cells are independently capable of driving neuroinflammation and could alter cognitive function when trafficked to specific areas of the CNS. 

Until recently, little was known about how peripheral immune cells potentiate cognitive decline and sickness behaviors during systemic inflammation. During advanced liver disease, microglia recruitment of monocytes through TNF-alpha signaling correlates with sickness behaviors and generalized neuronal excitability [121]. Burfeind and colleagues recently described a unique population of neutrophils that invade the brain in a murine pancreatic cancer cachexia model [117]. The authors identified a distinct population of neutrophils that specifically infiltrate the velum interpositum (VI), an area immediately adjacent to the hippocampus and habenula. When inflamed, the VI is uniquely positioned to drive cognitive decline and sickness behavior by virtue of its proximity to structures critical in memory and appetite regulation. Furthermore, the recruitment of this inflammatory population of neutrophils relies heavily on the CCR2-CCL2 axis, as genetic ablation of CCR2 reduces CNS neutrophil infiltration and ameliorates symptoms of cachexia. These findings present a novel mechanism by which peripheral tumors subjugate the brain and cause cognitive decline.

### 3.3. Tumor-Derived Extracellular Vesicles

Extracellular vesicles (EVs) are cell-derived membranous structures and include apoptotic bodies, microvesicles, and exosomes. Initially described as carriers of cellular waste [122], we now recognize EVs’ signaling potential through their exchange of lipids, nucleic acids, and protein products between cells [123]. Our knowledge of the molecular biology of EVs has expanded over recent years, shedding light on signaling functions of these extracellular molecules during both normal physiology and pathology. Recent studies demonstrate that tumor cells secrete EVs carrying oncogenic signaling molecules that are responsible, in part, for the pathogenesis and growth of cancer [124,125,126,127]. In the CNS, EVs shuttle cellular products important in synaptic plasticity, neurovascular permeability, and glial cell morphology [128]. Furthermore, it is now accepted that peripherally-derived EVs can readily cross the BBB both via endothelial transcytosis and through CVOs [129,130]. With the recent advances in EV biology, a thorough examination of mechanisms by which peripherally-derived EVs signal to the CNS may provide therapeutic insight for CRCI.

During cancer and other states of peripheral inflammation, a heterogeneous population of EVs containing inflammatory mediators are released into the circulation. One group of signaling molecules within EVs that have garnered attention in recent years is the inflammation-related microRNAs (miRs). EVs containing inflammatory miRs are significantly elevated in the serum of mice 1 hour after intraperitoneal injection of low dose lipopolysaccharide (LPS) [131]. When inflammatory EVs that are isolated from the serum of LPS-injected mice are reintroduced peripherally to a healthy animal, a distinct inflammatory response ensues within the brain. Specifically, hippocampal and neocortex microgliosis and astrocytosis are observed with concurrent increases in brain TNF-alpha and IL-6 [132]. These inflammatory changes within the brain correlate with EV levels of miR-155 and miR-146a within the cerebrospinal fluid (CSF) [132,133]. These studies demonstrate that peripheral EVs can cross the BBB and induce inflammatory responses in CNS structures important for cognition.

In a mouse model of peripheral inflammation induced by pancreatic ductal adenocarcinoma (PDAC), EVs containing miR-30 are up-regulated and abrogate tumor-suppressing mechanisms [134]. Interestingly, cerebral miR-30 directly impairs synaptic transmission and plasticity of hippocampal neurons [135]. The release of tumor-derived miR-30 into the circulation and subsequent localization to the CNS could provide an uninterrupted mechanism of tumor-induced cognitive impairment. During carcinogenesis, tumors that are more densely innervated are more aggressive and lead to increased mortality [136], and denervation of tumors in mouse models of PDAC slows the progression of the disease [137]. Consequently, it has been hypothesized that tumors signal to the peripheral nervous system to directly promote their innervation, a term called axonogenesis. The molecular mediators that control axonogenesis have remained poorly defined until recently. Indeed, tumor-derived exosomes containing EphrinB1 drive nervous system innervation of head and neck squamous cell tumors [138]. This study exemplifies how tumor-derived EVs exert systemic control of nervous tissue to dictate remodeling and signaling. Although not explored in the CNS, EV-driven axonogenesis and neuronal remodeling may prove to be important mechanisms for altering cognition during cancer progression. As our understanding of the basic biology of EVs continues to improve with the development of new technologies, identifying molecular targets of “inflammatory” EVs may provide therapeutic targets for cancer and its associated CNS sequelae.

### 3.4. Blood-Brain Barrier Integrity

The observation that water-soluble dyes injected systemically stain all organs except CNS structures provided the first piece of evidence that the brain is compartmentally distinct from the periphery [139]. These pioneering studies by Paul Ehrlich and his students demonstrated that a BBB exists during normal physiologic states. We now recognize the BBB as a critical gatekeeper preventing the unrestricted movement of micro and macromolecules to and from the CNS. Disruption of this barrier is generally accompanied by changes in CNS physiology. The BBB cellular components include endothelial cells, pericytes, and astrocytic end-feet, while an acellular basement membrane further divorces the luminal surface from the brain parenchyma. Disruption of any one component of the BBB exposes neurons to higher concentrations of circulatory molecules, which would otherwise remain tightly regulated under normal conditions. CNS exposure to higher circulating solute concentrations results in abnormal neuronal signaling and compromises synapse integrity, processes that may result in cognitive decline. Specifying whether BBB dysfunction results from or precedes neuroinflammatory insult is a difficult task. Consequently, little is known about how the BBB is affected by peripheral tumors. As reviewed below, recent murine models demonstrate that peripheral cytokine dysregulation, a common feature of non-CNS cancers, is sufficient to disrupt the BBB and induce cognitive decline.

BBB dysfunction is observed during several neurodegenerative and neurovascular conditions and is considered a driver of disease progression [140,141,142]. BBB breakdown is also observed during systemic inflammation, including states of endotoxemia induced by LPS [143,144] and polyinosinic-polycytidylic acid [145]. Models of colitis [146] and pancreatitis [147] also revealed unique alterations in BBB integrity. In a rat model of pancreatic encephalopathy, a severe complication of acute pancreatitis which presents with confusion and cognitive impairment, serum TNF-alpha and IL-6 significantly correlate with BBB permeability [147]. Indeed, TNF-alpha is partly responsible for the generation of reactive oxygen species, down-regulation of endothelial junction proteins, and induction of IL-6 in human brain microvascular endothelial cells [148,149]. Conversely, peripheral injection of the anti-inflammatory cytokine IL-10 lessens BBB attenuation during the progression of acute pancreatitis [150], demonstrating the necessity of chronic inflammatory cytokine production for attenuating the BBB. As discussed earlier, peripheral tumors produce inflammatory cytokines which enter the circulation and travel to the BBB. BBB disruption has been observed in response to TNF-alpha [151], IL-1B, IL-6 [152], and IFN-gamma [153], all cytokines are up-regulated during the progression of cancer. Although the BBB remains poorly studied in the context of cancer independent of treatment, it is plausible that these inflammatory cytokines play a dual role in attenuating the BBB and directly influencing neuronal plasticity during cancer. As our understanding of the BBB in the context of peripheral inflammation continues to grow, the development of therapies that adjust BBB permeability may be a useful therapeutic approach for mitigating cancer symptoms that are augmented in the CNS.

## 4. Conclusions and Future Directions

The chronic and compounding nature of inflammatory mediators created by peripheral tumors creates a formidable challenge for the CNS. Our knowledge of the associated symptoms during acute and chronic disease has boomed in recent years, and we now understand that the symptomology of disease profoundly impacts the survival of the organism. Cognitive alterations and sickness behaviors observed in acute disease primarily serve a protective role [154]. The principles of life history theory suggest that the shunting of energy from cognitive processes and general locomotion to the immune system aids in eliminating pathogens [155,156]. However, less is understood about cognitive decline and sickness behaviors during chronic non-infectious diseases, particularly cancer. Humans evolved to develop methods to survive bacterial and viral infections, rarely living long enough to develop tumors. We now understand that shunting of energy away from the brain and into survival functions is acutely beneficial, but detrimental during chronic diseases, such as cancer. This is exemplified during the pathogenesis of cancer cachexia, as the symptoms and metabolic aberrations exert devastating effects on quality of life, ability to tolerate treatment, and ultimate survival of the patient. While the classical sickness behaviors associated with cancer, including fatigue, anorexia, and lethargy, are well recognized as deleterious symptoms, only recently the clinical evidence for the cognitive decline has emerged as a negative predictor of survival. As a result, several biological questions related to CRCI remain unexplored, as the field has remained largely studied in a clinical context. In this article, we have reviewed recent clinical evidence of peripheral tumors affecting cognition and the putative biochemical mechanisms by which peripheral tumors directly subjugate the brain (Figure 1, Table 2).

Several challenges remain in defining, identifying, and measuring cognition in the clinic. Standardized assessments of cognitive decline in the context of cancer are lacking. Furthermore, proper assessment of cognitive decline requires knowledge of baseline cognitive performance and multiple follow-up visits. Assessing the biological mechanisms of CRCI remains a challenging, yet valuable endeavor of future investigation. A serious need exists for the development of murine models of CRCI. Specifically, a curable murine tumor model that exhibits cognitive decline and other sickness behaviors will aid in identifying tumor factors that cause pre-treatment cognitive decline as well as enduring memory impairment after the cancer is cured.

As the clinical course of CRCI becomes more apparent, our understanding of the biological mechanisms of cancer-CNS communication becomes ever more important. Identifying CRCI in the clinic remains challenging, and no unifying standardized exam for diagnosing CRCI exists. Ideally, detecting CRCI would entail both clinical examination and an objective imaging approach. Further elucidation of the mechanisms by which tumors cause cognitive decline will allow oncologists to tailor treatment regimens to mitigate CRCI, improve quality of life, and ultimately increase survival of future patients. 

## Figures and Tables

**Figure 1 cancers-11-00687-f001:**
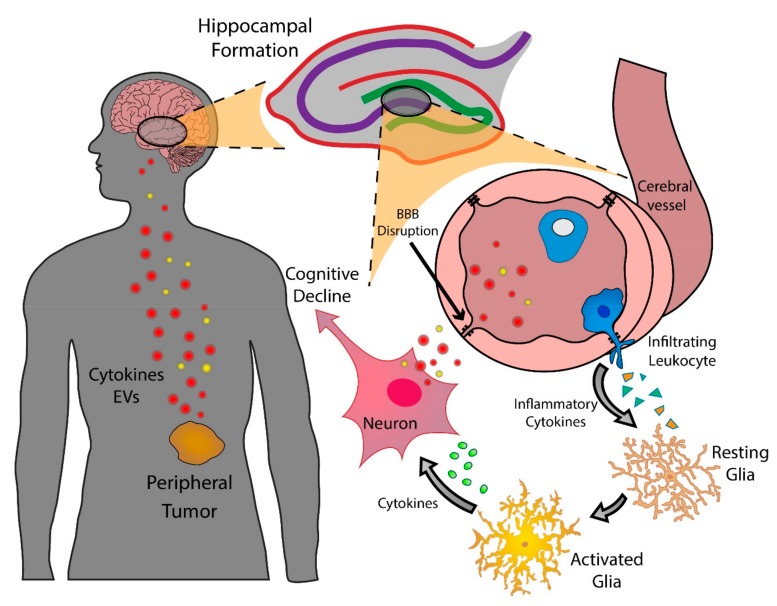
A model of putative mechanisms by which peripheral tumors interface with the CNS to initiate and sustain cognitive decline. EVs, extracellular vesicles; BBB, blood-brain barrier.

**Table 1 cancers-11-00687-t001:** Human studies reporting a significant cognitive decline in cancer patients prior to treatment.

Study	Cancer Type	Tested Cognitive Domains	Neuropsychological Assessments	% With Pre-Treatment Cognitive Impairment ^1^
Meyers et al. 2005	Acute Myelogenous Leukemia, Myelodysplastic Syndrome	Attention, motor function, memory, executive function, verbal fluency, visual-motor scanning speed, fine motor dexterity	Digit Span, Digit Symbol, HVLT, Controlled Oral Word Association, TMTA+B, Grooved Pegboard	>40%
Jansen et al. 2011	Breast	Attention, motor function, memory, executive function, visuospatial skill, language	RBANS, Stroop Test, Grooved Pegboard, AFI, CES-D Scale, STAI-S, LFS	23%
Vardy et al. 2015	Colorectal	Attention, memory, working memory capacity, task switching	Digit Span, Letter-Number, Spatial Span, Digit Symbol, TMTA+B, HVLT, Brief Visuospatial Memory Test	43%
Wefel et al. 2011	Nonseminamatous Testicular	Attention, motor function, memory, executive function, psychomotor speed, language	Digit Span, Digit Symbol, TMTA+B, MAE Controlled Oral Word Association, HVLT, Grooved Pegboard	46%
Baekelandt et al. 2016	Pancreatic	Self-reported assessment of cognitive function ^2^	EORTC QLQ-C30 Questionnaire	32%
Hsheih et al. 2018	Hematologic Cancers	Memory, executive function	Clock-in-the-Box (CIB), Five-Word Delayed Recall	>35%

^1^ Cognitive impairment as defined by significantly lower function in any one cognitive domain. ^2^ The raw scores from EORTC QLQ-30 on the two questions that together constitute the cognitive function scale were transformed to a score range from 0 to 100%. Low cognitive function was defined as a score <66.67%. Abbreviations: HVLT, Hopkins Verbal Learning Test; TMTA+B, Trail Making Test Part A + B; RBANS, Repeatable Battery of Adult Neuropsychological Status; AFI, Attentional Function Index; CES-D, Center for Epidemiological Studies-Depression; STAI-S, Spielberger State Anxiety Inventory; LFS, Lee Fatigue Scale; MAE, Multilingual Aphasia Examination; EORTC QLQ, European Organisation for Research and Treatment of Cancer Quality of life Questionnaire.

**Table 2 cancers-11-00687-t002:** Key mechanisms of the peripheral inflammation-CNS interface in cognitive decline.

Biological Target/Mechanism	Effect on CNS Function	References
**Cytokines**		
IL-1B	Modulation of neurotransmitter secretion, decrease in glutamatergic transmission	[76,77,78,79,80]
IL-2	Attenuation of Ach release, axonal degeneration, demyelination	[81,82,83]
IL-6	NMDA receptor neurotoxicity, microglial activation, impaired axonal regeneration	[90,91,93]
TNF-alpha	Promotes cerebral IL-1B production, MyD88 signaling, inflammatory astrocyte polarization	[98,99]
IFN-beta	Mediates cerebral endothelial release of CXCL10	[101]
**Immune cells**		
CD4+ T cells	Fas-Fas ligand-mediated dopaminergic toxicity	[112]
Neutrophils	NETosis, IL-17 secretion, cytotoxic granule spillage	[111,117]
**Extracellular vesicles**	Micro-RNA signaling, impairs synaptic transmission, axonogenesis	[135,138]
**Blood-brain barrier disruption**	Exposure of the brain to a higher concentration of peripheral solutes	[143,146,147]

Abbreviations: IL, interleukin; TNF, tumor necrosis factor; Ach, acetylcholine; NMDA, N-methyl-D-aspartate; MyD88, myeloid differentiation primary response protein 88; IFN, interferon; CXCL10, C-X-C motif ligand 10.

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
