# Peer review of "Pretreatment Cancer-Related Cognitive Impairment—Mechanisms and Outlook"

_cancers, 2019, doi:10.3390/cancers11050687_

Round 1
Reviewer 1 Report
The authors address the interesting issue of cognition in patients with cancer. This is a clinically relevant topic. The paper summarises well a developing field and does not over-sell the emerging story of cancer related cognitive impairment, but appropriately discusses potential mechanisms with appropriate evidence. More could be made of the parallel emerging story of inflammation in neurodegenerative diseases (eg https://www.ncbi.nlm.nih.gov/pubmed/25792098).
In the introduction it would be helpful to outline the definition and profile of cognitive impairment associated with cancer before diving in to other issues such as fatigue. This would help the reader to assess whether the domains of cognition affected might be affected by fatigue, depression or other issues. Perhaps moving the paragarph 91-100 further up would help achieve this?
The comment that fatigue is "analogous, yet biologically distinct from cognitive dysfunction" is difficult to understand. 'analagous' would suggest that cognitive dysfunction is indistinguishable from fatigue, which makes it difficult to see how they can then be considered distinct.
line 61/62 "the clinical course of fatigue is a notable
experience that impairs both mind and body, while CRCI is commonly
cryptic at clinical presentation" - this requires a bit more explanation. I don't quite understand how cognitive impairment would not be a notable experience by a patient. What does cryptic mean in this context? Usually it would mean it is hard to understand, or there is no cause.
Para 117-129. The authors should point out that the atrophy observed in these studies was not directly associated with cognitive impairment.
The authors discuss a range of potentially relevant mechanisms and use appropriate literature to back up their hypotheses.
The conclusion should outline some of the challenges faced in defining, identifying and measuring cognition. I would suggest this is vital before trying to develop a murine model, otherwise the model may not reflect the clinical disease.
Author Response
Reviewer 1:
Comments and Suggestions for Authors
The authors address the interesting issue of cognition in patients with cancer. This is a clinically relevant topic. The paper summarises well a developing field and does not over-sell the emerging story of cancer related cognitive impairment, but appropriately discusses potential mechanisms with appropriate evidence. More could be made of the parallel emerging story of inflammation in neurodegenerative diseases (eg https://www.ncbi.nlm.nih.gov/pubmed/25792098).
Discussion of this reference in relation to neuroinflammation caused by peripheral tumors is now included on page 5.
In the introduction it would be helpful to outline the definition and profile of cognitive impairment associated with cancer before diving in to other issues such as fatigue. This would help the reader to assess whether the domains of cognition affected might be affected by fatigue, depression or other issues. Perhaps moving the paragarph 91-100 further up would help achieve this?
Paragraph 91-100 has been moved to the introduction and precedes discussion of fatigue.
The comment that fatigue is "analogous, yet biologically distinct from cognitive dysfunction" is difficult to understand. 'analagous' would suggest that cognitive dysfunction is indistinguishable from fatigue, which makes it difficult to see how they can then be considered distinct.
The diction of this sentence has been changed to better contrast cognitive dysfunction from fatigue. Now reads as:
“Although out of the scope of this review, it is worth noting that cancer-related fatigue—defined as a persistent sense of tiredness related to cancer or cancer treatment that interferes with usual functioning—is biologically distinct from cognitive dysfunction.”
line 61/62 "the clinical course of fatigue is a notable experience that impairs both mind and body, while CRCI is commonly cryptic at clinical presentation" - this requires a bit more explanation. I don't quite understand how cognitive impairment would not be a notable experience by a patient. What does cryptic mean in this context? Usually it would mean it is hard to understand, or there is no cause.
The intent of this sentence was to emphasize that patients may not unanimously recognize their cognitive dysfunction, yet regularly recognize when they are fatigued. Furthermore, objectively defining cognitive decline requires a baseline understanding of patients’ cognitive function. We have added to this discussion to clarify this distinction. Now reads as:
“Additionally, the clinical course of fatigue is a notable experience that impairs both mind and body, while CRCI often insidious in nature, and patients may not universally recognize their cognitive decline during clinical presentation. Furthermore, defining cognitive decline requires knowledge of patients’ baseline cognitive function.”
Para 117-129. The authors should point out that the atrophy observed in these studies was not directly associated with cognitive impairment.
Mention of this is now included at the end of this paragraph.
The authors discuss a range of potentially relevant mechanisms and use appropriate literature to back up their hypotheses. The conclusion should outline some of the challenges faced in defining, identifying and measuring cognition. I would suggest this is vital before trying to develop a murine model, otherwise the model may not reflect the clinical disease.
Discussion of the challenges in assessing cognition has been added to the conclusion (lines 414-417).
Reviewer 2 Report
The manuscript deals with an important issue of cancer-related cognitive impairment.
• Page 2, lines 69-77: Reference 20 is dealing with patients at a mean age of 78.8 years. Cognitive impairment has been recently demonstrated in young patients with Hodgkin lymphoma (Trachtenberg et al, BJH, 2018). This issue should be also referred to in the review.
• Page 7, line 312: Please explain the abbreviation "CVO".
• Page 8, lines 336-339: The authors provide no data to this hypothesis and I am not sure that a speculation should be included in the review.
• Page 9, lines 384-385: Please provide data or a reference to this assumption.
• The authors may wish to consider adding a table presenting the essential mechanisms discussed with appropriate references.
• Please clarify reference # 30.
• Currently the manuscript is somewhat cumbersome and the ideas are not easy to grasp. I suggest excluding the hypotheses that are not based on data.
Author Response
The manuscript deals with an important issue of cancer-related cognitive impairment.
• Page 2, lines 69-77: Reference 20 is dealing with patients at a mean age of 78.8 years. Cognitive impairment has been recently demonstrated in young patients with Hodgkin lymphoma (Trachtenberg et al, BJH, 2018). This issue should be also referred to in the review.
Mention of this article in relation to CRCI in younger patients with Hodgkin lymphoma is now included in this paragraph.
• Page 7, line 312: Please explain the abbreviation "CVO".
This acronym (CVOs; circumventricular organs) is defined on page 5.
• Page 8, lines 336-339: The authors provide no data to this hypothesis and I am not sure that a speculation should be included in the review.
While we completely agree excessive speculation is not appropriate, we believe some expert opinion is expected in reviews of this type. Madeo et al elegantly demonstrate how tumor-derived EVs are capable of influencing peripheral nerve remodeling. We go on to say this interaction has not been explored in the context of the CNS, but may prove to be a mechanism of altered neurocircuitry influencing cognition.
• Page 9, lines 384-385: Please provide data or a reference to this assumption.
References have been added to provide rationale for sickness responses during infectious disease in the context of life history theory.
• The authors may wish to consider adding a table presenting the essential mechanisms discussed with appropriate references.
A table describing the key mechanisms and references has been added.
• Please clarify reference # 30.
Reference 30 is utilized to demonstrate cognitive impairment is present in patients with breast cancer even prior to treatment. This is demonstrated in breast cancer patients performing significantly worse on the single-item attention question and the RVP (Rapid Visual Processing) and TMT (Trail Making Test) tests.
• Currently the manuscript is somewhat cumbersome and the ideas are not easy to grasp. I suggest excluding the hypotheses that are not based on data.
Thank you for the thoughtful reviews on this paper. As we discuss in a previous comment, we sincerely believe the insight we provide that is not strictly backed by primary data is carefully worded and not overzealous. For example:
“Although not explored in the CNS, EV-driven axonogenesis and neuronal remodeling may prove to be important mechanisms for altering cognition during cancer progression.”
“Although the BBB remains poorly studied in the context of cancer independent of treatment, it is plausible that these inflammatory cytokines play a dual role in attenuating the BBB and directly influencing neuronal plasticity during cancer.”
Reviewer 3 Report
This review is well written and thorough - and deals with an important topic in cancer: cognitive impairment. However, the overall issue I have with the paper is that its title is a bit misleading/confusing - and perhaps a bit too general (all cancer related cognitive impairmenet). The primary focus of the paper purports to be about cancer related (mechanisms) cognitive impairment regardless of treatment status (pre-treatment, during treatment, post treatment). Page 2, line 90 seems to focus the review on pre-treatment CRCI but the rest of the paper is not so clear. Of course, there are significant challenges in determining cognitive impairment before cancer treatment but the authors do a good job in trying to summarize at least the observations. Correlations and interval change would be difficult to find literature on. Page 4 line 138 onwards, the authors delve into some complex pathophysiological/basic mechanisms. I think this section would benefit from a figure or at least a table to map out some of the proposed pathways. The current Figure 1 is good (well drawn too) but is specifically about the CNS interactions with tumors. The review would likely benefit from more visual exploration of the interleukins, TNF, interferon etc. Otherwise, the review is appropriate for Cancers given its comprehensive exploration of an important topic.
Author Response
This review is well written and thorough - and deals with an important topic in cancer: cognitive impairment. However, the overall issue I have with the paper is that its title is a bit misleading/confusing - and perhaps a bit too general (all cancer related cognitive impairmenet). The primary focus of the paper purports to be about cancer related (mechanisms) cognitive impairment regardless of treatment status (pre-treatment, during treatment, post treatment). Page 2, line 90 seems to focus the review on pre-treatment CRCI but the rest of the paper is not so clear. Of course, there are significant challenges in determining cognitive impairment before cancer treatment but the authors do a good job in trying to summarize at least the observations. Correlations and interval change would be difficult to find literature on.
The title now reflects that this review is focused on pre-treatment cognitive impairment in cancer.
Page 4 line 138 onwards, the authors delve into some complex pathophysiological/basic mechanisms. I think this section would benefit from a figure or at least a table to map out some of the proposed pathways. The current Figure 1 is good (well drawn too) but is specifically about the CNS interactions with tumors.
We have supplemented figure 1 with a table highlighting the key mechanisms and references.